# The Physiological Roles and Pathological Implications of Urea Transporters in the Cardiovascular System

**DOI:** 10.3390/biomedicines14010031

**Published:** 2025-12-23

**Authors:** Guangying Shao, Zhiwei Qiu, Min Li, Baoxue Yang, Xue Yu, Fusui Ji

**Affiliations:** 1Department of Cardiology, Beijing Hospital, National Center of Gerontology, Institute of Geriatric Medicine, Chinese Academy of Medical Sciences, Beijing 100730, China; shaoguangying5172@bjhmoh.cn; 2Institute of Clinical Pharmacology, Peking University First Hospital, Beijing 100034, China; 11067@pkufh.com; 3State Key Laboratory of Vascular Homeostasis and Remodeling, Department of Pharmacology, School of Basic Medical Sciences, Peking University, Beijing 100191, China; leemin@bjmu.edu.cn

**Keywords:** urea transporters (UTs), membrane channel proteins, cardiovascular diseases, urea

## Abstract

Urea transporter (UT) proteins are a group of membrane proteins specifically facilitating the transmembrane transport of urea, primarily divided into the UT-A and UT-B subfamilies. Early studies have predominantly focused on their pivotal roles in the mechanism of urine concentration in kidneys. Recently accumulating evidences suggest that UTs are also expressed in the cardiovascular system, particularly in cardiomyocytes and vascular endothelial cells, where they contribute to critical physiological processes such as regulation of cell volume homeostasis, modulation of nitric oxide production, control of myocardial electrophysiological properties, and adaptation to cardiac stress. Importantly, impairments or disruptions in UT activities have been increasingly associated with the pathogenesis and progression of multiple cardiovascular disorders, including hypertension, uremic cardiomyopathy, myocardial hypertrophy, heart failure, cardiac conduction disorders and atherosclerosis, which deepens the understanding of the role of urea metabolism as a key component in cardiovascular homeostasis. This brief review summarizes the distribution and physiological functions of UTs in the cardiovascular system, and evaluates the potential and existing challenges of targeting UTs as a novel therapeutic approach for cardiovascular diseases.

## 1. Introduction

The ammonia produced by amino acid catabolism is neurotoxic, and it is mainly converted into non-toxic urea in the liver through the highly conserved urea cycle. Urea, as the final product of amino acid metabolism, is released into the bloodstream and constitutes the primary source of plasma urea nitrogen. Owing to its low molecular weight and uncharged nature, urea can readily cross most cell membranes via passive diffusion [1]. However, its permeability through the lipid bilayer by passive diffusion is limited and insufficient to support the physiological demands for rapid transmembrane flux [2,3]. The rapid transmembrane transport of urea mainly relies on a specialized group of integral membrane proteins known as urea transporters (UTs), which can efficiently and selectively mediate the transmembrane transport of urea molecules. Studies have shown that the permeability of UTs can reach 10^4^ to 10^6^ urea molecules per second, which is 10 to 100 times higher than passive diffusion, thereby maintaining urea homeostasis and urine concentration under physiological conditions and other important physiological processes [4]. UTs are members of the solute carrier family 14 (SLC14) and mediate the passive transmembrane movement of urea in a concentration-dependent manner. This transport mechanism does not depend on sodium, chloride or other ions [5]. Urea is distributed throughout the body via the blood circulation, including the cardiovascular system. In blood vessels, urea can diffuse into endothelial cells, smooth muscle cells and their surrounding tissues. Under pathological conditions, the long-term elevated blood urea concentration will continuously expose vascular wall cells to a high urea environment, which may interfere with cell functions [6].

Basing on gene coding and structural features, mammalian UTs can be classified into two subfamilies, UT-A and UT-B [7,8]. Recent research has utilized cryo-electron microscopy to determine the high-resolution structures of UTs, uncovering the homotrimeric architectures of both UT-A and UT-B. In particular, Huang et al. have reported the structure of human UT-B in the resting conformation, demonstrating that the protein assembles into parallel trimers embedded in the membrane, with a prominent central cavity formed at the trimeric interface (Figure 1) [4]. These structures exhibit a conserved structural motif characterized by two homologous domains that enclose a continuous transmembrane channel, constituting a solvent-accessible pore for urea transport [8,9]. Some small urea analogs, such as dimethylthiourea (DMTU), have been reported as non-selective inhibitors of UT-A and UT-B through competitive binding. Based on the progress of high-throughput screening and UTs high-resolution structural analysis technology, specific inhibitors for UTs have been continuously developed [10,11,12].

The UT-A subfamily consists of six members (UT-A1 to UT-A6), encoded by the *SLC14A2* gene, and generated through alternative promoters utilization and differential mRNA splicing. UT-A1 is the largest isoform in this subfamily in terms of structure and is predominantly localized to the apical membrane of the inner medullary collecting duct (IMCD). UT-A2 shares an identical C-terminal domain with UT-A1 and is distributed in the thin descending limb (TDL) of the loop of Henle. UT-A3 shares the N-terminal sequence with UT-A1 but terminates at exon 13 and is expressed on the basolateral membrane of the IMCD. UT-A4 has been mainly detected in the renal medulla of rats, UT-A5 is specifically expressed in the testis of mice, and UT-A6 has been found in the human colon tissue [13,14,15,16,17]. The UT-B subfamily is encoded by the *SLC14A1* gene, which is widely distributed in various tissues, including red blood cells, endothelial cells of the descending vasa recta (DVR) in the kidney, brain, heart, spleen, testis, bladder, and vascular endothelium (Figure 2) [9,18]. This tissue-specific and diverse distribute indicates that UTs may perform specific urea transport functions in different physiological environments, providing a structural basis for understanding the role of urea in maintaining metabolic homeostasis.

The core physiological function of UTs is to mediate urea transmembrane transport, thereby participating in maintaining the systemic urea homeostasis and osmotic pressure balance. In the liver, the urea cycle mediates the conversion of neurotoxic ammonia into non-toxic urea, a critical process for maintaining whole-body nitrogen homeostasis [19]. As a small, membrane-permeable solute, urea contributes significantly to the effective osmotic pressure across tissues. Through its dynamic distribution between cells and tissues, it directly participates in and precisely regulates water and salt balance as well as cell volume [20,21]. Especially in renal urine concentration, urea works with water and sodium reabsorption along the nephron to establish and maintain the increasing osmotic gradient from the outer to inner medulla, enabling the formation of concentrated urine [22,23]. In the kidney, different UT subtypes work in concert to facilitate the urine concentration process. UT-A1 and UT-A3 are responsible for the trans-epithelial absorption of urea in the IMCD. UT-A2 is involved in urea recycling in the TDL of the loop of Henle, and UT-B regulates urea exchange between the DVR and the renal medullary interstitium [24,25]. This spatially coordinated urea recycling system is of great significance for establishing and maintaining the hypertonic medullary environment, which drives efficient water reabsorption.

Besides the kidney, UTs also exert various functions in other tissues. For instance, UT-B maintains intracellular urea homeostasis in erythrocytes, may influence spermatogenesis in the testis, and potentially modulates cardiac electrophysiological activity [9]. The activity of UTs is subject to multi-level regulation, including both short-term and long-term mechanisms [26]. Short-term regulation mainly involves post-translational modifications, such as phosphorylation, which can alter their transporter trafficking and membrane activity. Long-term regulation includes adaptive changes in gene expression levels, such as dietary protein intake can modulate the expression of renal UT-A subtypes, a low-protein diet can upregulate UT-A2, antidiuretic hormone (ADH) can also regulate the expression and function of UTs [27,28,29]. These regulatory mechanisms work together to enable the precise control of urea transport, enabling the body to sustain relative stability in urea metabolism and osmotic pressure under different physiological conditions.

For a long time, urea has been employed as a clinical biomarker for the assessment of renal function. Since the clearance of urea almost entirely relies on the glomerular filtration of the kidneys and partial reabsorption by the renal tubules [30]. The homeostasis of its blood concentration highly depends on the glomerular filtration rate (GFR) and the integrity of renal tubular function. Therefore, elevated urea levels may serve as an early and sensitive marker of renal dysfunction [31]. However, accumulating evidence now demonstrates that elevated urea—observed in chronic kidney disease (CKD), and even at the upper limit of the normal range, significantly promoting cellular dysfunction and injury in organs outside the kidneys, particularly in the cardiovascular system [32,33,34]. This concept redefines urea as a true “uremic toxin”, with the cardiovascular system representing a principal target of its destructive effects.

The mechanisms by which urea induces cardiovascular damage are complex and multifactorial. Elevated urea concentrations induce endothelial dysfunction, which is a critical early event in the development of cardiovascular disorders such as atherosclerosis and hypertension [35,36]. Studies have confirmed that urea affects the nitric oxide (NO) signaling pathway in endothelial cells, resulting in a series of functional disorders including changed vasodilation, increased inflammatory responses, and enhanced leukocyte adhesion [37,38]. Furthermore, high urea levels exert direct cardiotoxic effects by suppressing cardiomyocyte contractility, inducing mitochondrial oxidative stress, and activating apoptotic and pro-fibrotic signaling cascades, all of which drive myocardial remodeling and functional decline [39,40]. The direct inhibition of cardiac ion channels by urea slows down the conduction velocity, increasing the susceptibility to conduction blocks and arrhythmias [40]. Moreover, urea can also aggravate systemic inflammation and vascular damage by promoting protein carbamylation, inducing pro-inflammatory, pro-atherosclerotic and pro-apoptotic factors [41,42].

As a highly polar molecule, urea achieves transmembrane transport into cardiovascular cells (such as endothelial cells and cardiomyocytes) through UTs. This review summarizes the potential roles of UTs in the occurrence and progression of various cardiovascular diseases. Clarifying these mechanisms not only offers novel insights into the pathophysiological basis of cardiovascular disorders but also establishes a robust theoretical framework for developing innovative therapeutic strategies that target UTs.

## 2. Hypertension

The pathophysiology of hypertension involves multiple mechanisms, including dysregulation of vascular tension and disorders in water and salt homeostasis [43]. Diuretics are widely used as first-line therapeutic agents in the clinical management of hypertension [44]. The main mechanism involves enhancing sodium ion excretion, increasing urinary osmotic pressure, and promoting water excretion, thereby reducing water and sodium retention and lowering blood volume. However, long-term or improper use of traditional diuretics is often accompanied by a series of adverse reactions, including electrolyte imbalances, elevated blood uric acid and abnormal glucose metabolism [45,46]. These limitations have driven the search for new regulatory pathways and therapeutic targets for blood pressure control. Notably, emerging evidence has uncovered critical roles of UTs in the cardiovascular system, particularly in the modulation of blood pressure, providing a new research direction for the development of innovative antihypertensive therapies [11].

Individuals with genetic loss of UT-B (Kidd antigen) are unable to concentrate urine beyond 800 mOsm/kg H_2_O [47]. Similarly, UT-B knockout mice exhibit a mild reduction in urinary concentrating capacity [48,49]. UT-A2 knockout mice also display significant impairments in this function [50]. All-UT-knockout mice exhibit polyuria and a significant reduction in urine osmotic pressure. They are unable to effectively concentrate urine under stress conditions such as dehydration, acute urea load or high protein intake, revealing the crucial role of UT proteins in maintaining the water and salt balance [48,51].

NO is a critical mediator of vascular homeostasis and endothelium-dependent vasodilation and is generated from substrate L-arginine via endothelial nitric oxide synthase (eNOS). Urea is closely related to the NO metabolic pathway and contributes to the regulation of vascular tension [52,53,54]. The pathophysiological mechanism is strongly supported by evidence from UT-B gene knockout mouse models, which exhibits reduced baseline blood pressure, markedly enhanced vasodilatory responses to acetylcholine, increased phosphorylation of eNOS, and downregulated arginase I expression in vascular tissues, thus promoting the biosynthesis of NO (Figure 3) [55]. In addition, specific UT-B inhibitors can induce vasodilation in a concentration-dependent manner and significantly lower blood pressure in spontaneously hypertensive rats, which can be abolished by NOS inhibitors, confirming its dependence on the NO signaling pathway [55,56]. Notably, in UT-B knockout mice, there was an increase in plasma prostacyclin levels and a decrease in plasma angiotensin II levels. Given that the cyclooxygenase-prostacyclin pathway can also promote vasodilation, UT-B may be involved in the reduction in blood pressure in mice through mechanisms beyond the eNOS-NO pathway [55]. Furthermore, UT-B contributes to the countercurrent exchange of urea from the ascending to descending vasa recta, thereby maintaining a high urea concentration in the renal inner medulla [7]. This process can influence the urine concentration capacity and long-term blood pressure. However, current studies predominantly employ UT-B knockout mice or UT-B inhibitors. Given the widespread expression of UT-B, identifying the precise cellular source responsible for modulating hypertension progression remains challenging. Therefore, it is necessary to construct conditional knockout mice in the future for in-depth verification, which can further clarify the tissue specificity of UT-B.

The UT-A1/3 knockout mice are characterized by sustained polyuria, which is caused by osmotic diuresis due to high urea. When dietary protein intake was restricted and urea production was reduced, this polyuria phenomenon was significantly alleviated [57]. Duchesne et al. also demonstrated that UT-A1 is involved in compensatory regulation during hypertension. In an angiotensin II-induced hypertension model, the expression of UT-A1 in the renal medulla was significantly downregulated, accompanied by changes in the expression of other transporters [56]. These coordinated changes promoted diuresis and natriuresis, thereby contributing to the mitigation of elevated blood pressure. Furthermore, in kidney, pharmacological or genetic inhibition of UT-A1 can block urea reabsorption in the medullary collecting ducts and the loops of Henle, leading to osmotic diuresis and indirectly modulating long-term blood pressure homeostasis [58,59]. Genome-wide association studies (GWAS) have further identified polymorphisms in the human *SLC14A2* gene, encoding UT-A, as being significantly associated with blood pressure levels and hypertension risk, providing robust genetic evidence for the involvement of UTs in human cardiovascular physiology. Specifically, two single nucleotide polymorphisms (SNPs), Val227Ile (rs1123617) and Ala357Thr (rs3745009), were significantly associated with interindividual blood pressure variability in males [60]. Importantly, these genetic variants may also influence the therapeutic response to antihypertensive drugs. Hong et al. reported that among patients treated with the gastrointestinal therapeutic system (GITS) of nifedipine for antihypertensive control, individuals homozygous for the Ala357 allele (Ala357/Ala357) at the Ala357Thr locus and those carrying either the Val227/Ile227 or Ile227/Ile227 genotype at the Val227Ile locus exhibited the greatest reductions in both systolic and diastolic blood pressure [61,62].

Based on these mechanistic insights, UT inhibitors have been expected to be promising candidates for a novel class of antihypertensive agents. In contrast to traditional diuretics, UT inhibitors offer distinct advantages: they not only induce diuretic effects by blocking the reabsorption of urea in kidney, but also promote vasodilation through the modulation of vascular endothelial function. Crucially, UT inhibitors exert their diuretic effects without inducing significant electrolyte imbalances [63,64,65]. These preclinical findings collectively indicate that UT inhibitors can achieve blood pressure reduction through dual mechanisms—synergistic improvement of vascular function and enhanced water excretion, and thus expected to be a new class of antihypertensive drugs, particularly in patient populations at high risk for electrolyte disturbances or those with underlying endothelial dysfunction.

## 3. Uremic Cardiomyopathy

Uremic cardiomyopathy is a common cardiovascular complication in patients with CKD, characterized by pathological alterations including left ventricular hypertrophy, myocardial fibrosis, and impaired cardiac diastolic function. This specific type of myocardial injury is primarily driven by multiple interrelated factors associated with CKD, such as sustained hypertension, volume overload, overactivation of the renin-angiotensin system (RAS), and the accumulation of uremic toxins [66,67,68]. Although angiotensin-converting enzyme inhibitors (ACEIs) and loop diuretics are mainly used for intervention in clinical practice at present, their therapeutic efficacy is often limited, and they may induce adverse effects such as electrolyte disturbances [69,70,71].

Urea plays a critical role in the pathogenesis and progression of uremic cardiomyopathy. As renal function declines, plasma urea level progressively rises. Excessive urea can enter cardiomyocytes via specific UTs, directly triggering myocardial remodeling and fibrotic changes [72,73]. UT-A subtypes are distributed in both cardiomyocytes and cardiac fibroblasts, implicating they may participate in the process of urea-induced myocardial injury [56,74,75]. In a murine model of uremic cardiomyopathy induced by 5/6 nephrectomy, Kuma et al. demonstrated the expression level of total UTs in cardiac tissue was elevated and positively correlated with the degree of myocardial fibrosis. Overexpression of UTs in cultured H9c2 cells (cardiac myoblasts) significantly increased the amount of vimentin in the cells, suggesting that UTs may serve as key mediators in uremic cardiomyopathy. Following 5/6 nephrectomy, compared with wild-type CKD mice, UT-A1/A3 knockout mice exhibited significantly attenuated myocardial fibrosis and hypertrophy. Furthermore, treatment of CKD mice with the UT inhibitor DMTU not only effectively lowered blood pressure but also substantially ameliorated cardiac hypertrophy and fibrosis. UT inhibition led to a significant reduction in the myocardial mRNA expression of local renin and angiotensin-converting enzyme (ACE), as well as decreased expression of fibrotic markers, including α-smooth muscle actin (α-SMA) and vimentin. These findings suggest that inhibition or deletion of UTs suppresses hypertension and attenuates cardiac hypertrophy and fibrosis in CKD-induced uremic cardiomyopathy mice [76].

Moreover, DMTU has been shown to exert cardioprotective effects in models of myocardial infarction and diabetic cardiomyopathy, suggesting that its protective mechanisms may be universal and extend beyond uremic conditions [77]. Based on the aforementioned research evidence, targeting UTs may represent an attractive therapeutic option for preventing uremic cardiomyopathy and other fibrosis-related disorders.

## 4. Myocardial Hypertrophy and Heart Failure

Myocardial hypertrophy and heart failure are the common final pathways in the progression of diverse cardiovascular diseases, characterized by complex pathophysiological mechanisms and limited therapeutic efficacy [78,79]. Accumulating studies have indicated that dysregulated urea metabolism and its transporter system play a critical role in myocardial remodeling.

Studies have demonstrated that three UT-A subtypes with molecular weights of 56 kDa, 51 kDa, and 39 kDa are expressed in rat hearts, while four UT-A proteins subtypes with molecular weights of 97 kDa, 56 kDa, 51 kDa, and 39 kDa are expressed in human hearts. In experimental models such as uremic rats, DOCA/salt-induced hypertension model, and angiotensin II infusion model, the 56 kDa UT-A glycoprotein was markedly upregulated. Notably, in the heart tissues from patients with end-stage heart failure, both 56 kDa and 51 kDa UT-A proteins exhibited significantly increased expression. These results suggest that UT-A may serve as an important biomarker for myocardial remodeling, but the specific subtypes and corresponding cell types need to be further confirmed [56,80].

At the metabolic level, urea modulates cardiomyocytes function via the polyamine synthesis pathway [81,82]. Under the catalysis of arginase, arginine is hydrolyzed into ornithine and urea. The generated ornithine then enters the polyamine (putrescine, spermidine, spermine) synthesis pathway driven by the rate-limiting enzyme ornithine decarboxylase (ODC) [83]. Studies have shown that pressure overload can upregulate the activities of arginase II and ODC in cardiomyocytes significantly, jointly driving the accumulation of polyamines, which in turn promotes protein synthesis, cardiomyocyte hypertrophy and pathological gene reprogramming [84,85]. In models of myocardial hypertrophy, the activity of key enzymes in this pathway—particularly ODC—was significantly elevated, and pharmacological inhibition of these enzymes effectively attenuated hypertrophic progression [86]. Therefore, the increase in urea levels may indicate an enhanced flow of arginine metabolism towards the hypertrophy-promoting polyamine pathway.

In the UT-B deficiency model, Duchesne et al. demonstrated through proteomic analysis that UT-B knockout mice exhibited downregulation of multiple mitochondrial respiratory chain proteins, leading to impaired electron transport chain function, a marked increase in reactive oxygen species (ROS) production, reduced mitochondrial membrane potential, and disrupted ATP synthesis. These mitochondrial abnormalities collectively impair cardiac adaptation to stress overload, promoting spontaneous myocardial hypertrophy and structural and functional cardiac remodeling [56,87]. Furthermore, Du et al. reported that UT-B knockout mice were susceptible to cardiac oxidative stress and myocardial hypertrophy, accompanied by elevated intracellular urea concentrations and disturbances in glucose and lipid metabolism [88].

Based on these findings, future research should prioritize elucidating the subtype-specific functions of UTs in myocardial tissue and investigating their interactions with key signaling pathways, which involved in myocardial hypertrophy and heart failure. These studies will not only contribute to deepen our understanding of the underlying pathophysiological mechanisms of heart failure but may also bring new inspirations to novel therapeutic interventions against this major public health challenge.

## 5. Cardiac Conduction Disorders

Cardiac conduction disorders represent a critical pathological process in cardiovascular diseases, characterized by delayed or disrupted propagation of electrical impulses within the heart’s specialized conduction system. Clinically, these disorders commonly present as atrioventricular conduction block, which can lead to syncope or sudden cardiac death in severe cases. Traditionally, the underlying causes have been primarily linked to myocardial ischemia, fibrotic remodeling, inflammatory processes or inherited ion channelopathies [89,90].

In recent years, UT-B, an atypical protein expressed in cardiac tissues, has emerged as a novel factor implicated in conduction disturbances upon functional deficiency. UT-B is specifically distributed in the heart, and its role in cardiac conduction was first elucidated in the UT-B knockout mice. Consistent findings have demonstrated that from juvenile (6 weeks old) to elderly (52 weeks old), UT-B knockout mice exhibited a significant prolongation of the P-R interval on electrocardiography (ECG), indicating delayed atrioventricular conduction. Importantly, the conduction defect progressed in an age-dependent manner, with approximately 20% of elderly UT-B knockout mice developing grade II or III atrioventricular block. Electrophysiological recordings showed reduced action potential amplitude (APA) and upstroke velocity (Vmax) in cardiomyocytes from UT-B knockout mice, indicating impaired excitability and conduction [87].

Since UT-B is mainly involved in urea transport, this apparent electrophysiological abnormalities raise a key question: what mechanisms underlie the link between UT-B deficiency and cardiac conduction dysfunction? Evidence suggests intracellular urea accumulation and its downstream molecular and cellular effects. Lv et al. have proposed that urea accumulation caused by UT-B deficiency may promote atrioventricular block by triggering inflammatory pathways [91]. Bioinformatic analyses have further implicated regulatory networks such as the ceRNA axis (NONMMUT140591.1–mmu-miR-298-5p–Gata5) in the molecular pathogenesis of conduction disturbances following UT-B knockout [92]. Proteomic profiling of cardiac tissue from UT-B knockout mice has revealed dynamic alterations in proteins involved in cardiac contractility, energy metabolism, ion channel function, and oxidative stress responses [93]. Notably, changes in troponin T2 (TNNT2) and atrial natriuretic peptide (ANP) were particularly notable [94]. TNNT2, a core component of the cardiac troponin complex and a key calcium sensor, directly modulates myocardial contractility [95]. It was found that the expressions of TNNT2 were significantly upregulated in 16-week-old UT-B knockout mice without severe conduction block. However, TNNT2 levels were downregulated in 52-week-old mice exhibiting grade II–III block, similar to that of young wild-type controls. This dynamic regulation suggests a potential role of TNNT2 in the early compensatory and late decompensatory processes of cardiac conduction defects. In contrast, ANP, as a well-established biomarker and diagnostic hallmark of cardiac hypertrophy, was specifically upregulated in the hearts of 52-week-old UT-B knockout mice with conduction block, but remained unchanged in young mutant mice and age-matched wild-type mice. The results imply a strong association between ANP elevation and the transition to severe conduction impairment, suggesting that pathological cardiac hypertrophy may act as the role of “the last straw” in the deterioration of electrical conduction [91,96,97].

In addition to the aforementioned molecular mechanisms, severe mitochondrial dysfunction has also emerged as a central mechanism underlying cardiac conduction impairment in UT-B knockout mice. Du et al. have demonstrated that the expressions of 15 proteins in respiratory chain complexes I, III, IV, and V were significantly downregulated in myocardial mitochondria, leading to markedly impaired electron transport chain activity. The mitochondrial deficiency resulted in a reduction in mitochondrial membrane potential (ΔΨm), diminished ATP synthesis, and excessive production of reactive oxygen species (ROS). The study further confirmed that UT-B knockout mice suffered from severe oxidative stress damage, which provided an important cellular pathological foundation for the development of conduction block [88].

These findings provide novel perspectives for the understanding of the pathophysiology of cardiovascular diseases. In the occurrence and development of cardiac conduction disorders, regulating the function of cardiac UT-B or correcting its downstream molecular and cellular consequences could help prevent or delay the onset of conduction abnormalities. Targeting antioxidant therapies, such as the use of Mito-TEMPO and other mitochondria-specific antioxidants, has demonstrated cardioprotective effects in various cardiomyopathy models [98]. Finally, the dynamic change patterns of ANP and TNNT2 in UT-B deficiency-related cardiac conduction disorders suggest the potential as combined biomarkers for clinical risk assessment. The changes in the three over the course of the disease still need to be validated in clinical practice, thereby promising the identification of high-risk individuals and early intervention before irreversible structural remodeling.

## 6. Atherosclerosis and Endothelial Dysfunction

Atherosclerosis is a chronic inflammatory vascular disease. Endothelial dysfunction, as the initiating step, is characterized by a reduced bioavailability of NO, which leads to abnormal vascular dilation, adhesion of inflammatory cells and proliferation of smooth muscle cells [99,100,101]. Recent studies have found that UT-B is widely expressed in vascular endothelial cells and participates in the maintenance of vascular homeostasis by regulating urea permeability.

The accumulation of urea impairs endothelial function through multiple mechanisms. One of the relevant mechanisms is two competing metabolic pathways for L-arginine [102]. Inhibition of UT-B can upregulate the L-arginine-eNOS-NO pathway and downregulate the L-arginine–arginase–urea pathway. NO release was increased, promoting vasodilation. Gambardella et al. proposed a new concept that L-arginine is transported into endothelial cells through membrane transport proteins, with at least 80% of the transport occurring through the cationic amino acid transporter family. Under the high concentration of urea in uremia, urea can inhibit other membrane transport processes, thereby inhibiting the transport of L-arginine, ultimately reducing the synthesis of NO. The UT inhibitor phlorizin eliminated the inhibitory effect of urea on the transport of L-arginine [103]. When urea transport was inhibited, high levels of extracellular urea had no effect on L-arginine transport in vitro. High levels of urea do not affect the expression of UT-B, indicating that the effect of high urea on UT-B in endothelial cells is functional [55,101,104,105,106].

Additionally, a high urea environment promotes the carbamylation modification of proteins, a post-translational modification that generates pro-inflammatory molecules such as carbamylated low-density lipoprotein (cLDL). Then cLDL are phagocytosed by macrophages through scavenger receptors, accelerating foam cell formation and directly exacerbating the development of atherosclerotic plaques [107,108]. Moreover, Verbrugge et al. found that cyanate may induce protein carbamylation and damage endothelial function, thereby potentially increasing cardiovascular risk, whether generated during uremia-associated renal insufficiency or at sites of local vascular inflammation [109].

By specifically relieving the competitive inhibition of UT-B on L-arginine uptake, thereby restoring NO bioavailability and endothelial function, the initiation and progression of atherosclerosis can be effectively prevented at an early stage. But it is worth noting that in addition to urea, UT-B also transports water, ammonia, and various urea analogs, including formamide, acetamide, methylurea, methylformamide, thiourea, acrylamide and carbonate, while UT-A-mediated transport is relatively specific [110,111]. Whether UT-B plays the role solely through urea metabolism permeability in atherosclerosis and endothelial dysfunction needs to be further verified by in vitro and in vivo experiments, to assess the potential of pharmacologic modulation of vascular UT-B as a therapeutic strategy.

## 7. Summary

This review summarizes the emerging possible roles of UTs in cardiovascular pathophysiology through a comprehensive synthesis of current evidence (Table 1). Accumulating studies indicate that urea transport mediated by UTs extends beyond their classical function in osmotic balance and may actively contribute to the regulation of cardiovascular homeostasis. This shift in understanding prompts us to evaluate the pathophysiological significance of urea and its transporter systems in cardiovascular diseases [112,113]. Specifically, UT-A, predominantly expressed in the kidneys, may exert indirect cardiovascular effects by modulating systemic urea levels, whereas the expression of UT-B in the heart and vascular endothelial cells points to a potential for direct and localized regulatory roles in the cardiovascular system [114].

The urea-dependent efficient urine concentration mechanism not only maintains water and sodium balance but also participates in regulating effective circulating blood volume and cardiac preload. When effective blood volume is relatively insufficient, it will continuously activate the RAS and the sympathetic nervous system, leading to vasoconstriction and sodium retention, and jointly leading to blood pressure elevation [115]. In the long term, the dysfunction of urine concentration accompanied by abnormal urea metabolism is not only related to hypertension but also directly participates in the progression of left ventricular hypertrophy, vascular remodeling, and endothelial dysfunction through the continuous activation of neuroendocrine mechanisms, ultimately increasing the risk of heart failure and major adverse cardiovascular events [116,117]. Therefore, clarifying the precise mechanism of urea and specific UTs in integrating renal urine concentration function and overall cardiovascular regulation will provide a key theoretical basis for in-depth understanding of the pathological basis of cardiorenal comorbidity and the development of targeted intervention strategies.

Current evidence indicates that the impaired function or abnormal expression of UT-B may be associated with pathological features such as energy metabolism disorders in cardiomyocytes, elevated oxidative stress levels, and delayed electrical conduction. In terms of vascular function, the competitive relationship of arginine metabolic pathways provides a possible mechanism framework for understanding the connection between urea metabolism and endothelial dysfunction, indicating that UT-B may indirectly regulate the synthesis of NO by influencing the availability of L-arginine. This point seems not entirely consistent with the view that excessive urea in cells inhibits arginase activity, thereby promoting the competitive binding of NOS with L-arginine and increasing NO production. This may also reveal the complex interaction between UT-B and the arginine-NO pathway. Under physiological conditions, UT-B fine-tunes intracellular urea concentrations, potentially triggering compensatory downregulation of arginase in vascular tissues. This metabolic reprogramming shifts arginine flux toward the nitric oxide synthase (NOS) pathway, thereby supporting the maintenance of basal vascular tone. In contrast, under uremic conditions, excessive extracellular urea enters cells via UT-B, leading to impaired L-arginine transport and reduced NO synthesis. In this context, UT-B transitions from a homeostatic regulator to a conduit facilitating pathological urea influx. Consequently, pharmacological inhibition of UT-B can mitigate these detrimental effects. However, which pathway plays a more significant role still requires further verification in different disease contexts in the future [55]. These findings collectively highlight that the UT family, especially UT-B, may serve as an important node linking nitrogen metabolism and cardiovascular homeostasis, and UT-B dysfunction could represent a potential factor in the progression of certain cardiovascular diseases. These findings suggest that the role of UT-B within the arginine-NO axis is context-dependent, and the final phenotype of UT-B function is determined by the context dependency of its role, which is influenced by the differences in research models (gene knockout and pharmacological inhibition), disease stages (early compensation and late decompensation), and system background (local vascular and systemic metabolism). This requires the development of more refined research tools (such as tissue-specific regulatory models and dynamic metabolic tracking), and design of targeted strategies that can selectively correct the harmful functions of UT-B in pathological conditions without interfering with its physiological homeostatic role. In-depth exploration of this issue will provide critical insights for developing targeted metabolic therapies in cardiorenal syndrome.

Advances in this field have brought about new ideas for the prevention and treatment of cardiovascular diseases, which can drive the transformation of cardiovascular disease prevention and treatment strategies towards targeting the metabolic microenvironment. UTs represent a potential class of new drug intervention targets [10,64]. As a new class of potential drug targets, the translational value of UTs may not only lie in regulating urea itself, but also in their role as “metabolic homeostasis regulators” that may restore impaired communication between cells and organs. Theoretically, designing specific inhibitors for different subtypes and tissue distributions may achieve differentiated therapeutic goals. For instance, renal UT-A inhibitors may be employed to manage volume load and urea levels, while UT-B modulators targeting vascular endothelium may offer a new approach to improving endothelial function. Therefore, future drug development needs to go beyond the traditional “inhibition/activation” mindset, considering tissue selectivity and disease state dependence, in order to reduce the systemic urea load while precisely correcting metabolic-signaling disorders in specific cellular compartments (such as endothelial cells). However, realizing this vision faces multiple challenges: at the basic level, it is necessary to further identify the conformational dynamics and allosteric regulation mechanisms of UTs in different microenvironments with the aid of cryo-electron microscopy and computational simulations; at the translational level, it is necessary to establish preclinical models that can simulate the complex processes of human heart and kidney diseases, and explore the synergistic strategies of combining UTs regulators with existing therapies.

Nevertheless, it is crucial to acknowledge the current limitations of this research area. The primary limitation lies in that most mechanistic evidence derives from transgenic animal models. While these models are valuable for uncovering potential biological pathways, they have different pathophysiological processes from the complex and long-term progression of human chronic diseases. Therefore, the precise contribution of UTs to human cardiovascular pathology remains to be fully elucidated. Secondly, the relative contributions, regulatory mechanisms, and interactions of UT-A and UT-B in different cardiovascular cell types remain very poorly understood. Additionally, it remains unclear whether the activity of UTs is subject to feedback regulation by cardiovascular diseases. Furthermore, recent studies have revealed that UT-B facilitates the transport of various substances beyond urea. However, whether these substrates compete for transport and whether they participate in the pathogenesis and progression of related diseases remain to be fully elucidated [110,118].

Therefore, future related studies can be deepened along the following directions: First, it is necessary to verify existing findings in models more accurately reflect human physiology and pathology, such as tissue organoids, disease-specific iPSC-derived cells, to promote the preclinical development and safety evaluation of highly selective UT modulators. Second, proteomics and single-cell sequencing tools should be employed to precisely characterize UT-interacting proteins and downstream signaling networks in the cardiovascular system, and clarify their specific roles in specific pathological processes. Third, to establish a mechanistic foundation for future precise intervention, studies are needed to investigate the associations between UT gene polymorphisms, dynamic urea fluctuation patterns, and specific cardiovascular phenotypes, such as cardiac conduction disorders and microvascular dysfunction.

In conclusion, a deeper investigation into the functional roles of UTs may not only expand our understanding of the dimensions of cardiovascular metabolic regulation but also provide scientifically valuable insights for the development of novel therapeutic strategies. However, the transition from theoretical concepts to clinical applications still requires systematic and rigorous follow-up research through multi-disciplinary collaboration.

## Figures and Tables

**Figure 1 biomedicines-14-00031-f001:**
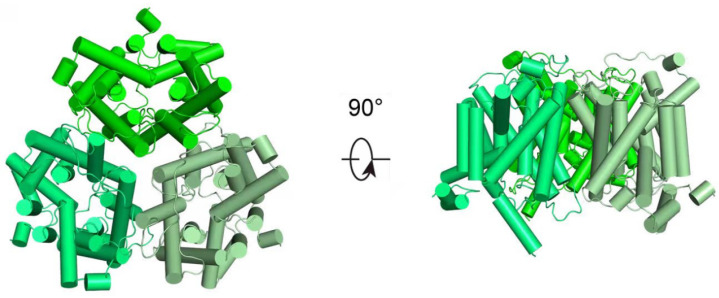
Structural representation of hUT-B. Structural representation of apo-hUT-B homotrimer, from the extracellular view (**left**) and the side view (**right**), respectively.

**Figure 2 biomedicines-14-00031-f002:**
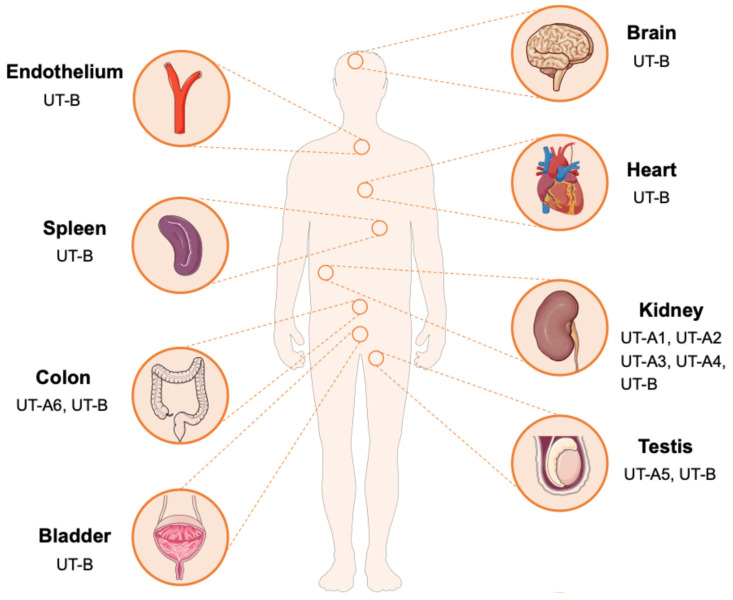
Tissue distribution of UTs. UT-A1-A4 subtypes are mainly distributed in the kidney. UT-A5 is distributed in the testicles, and UT-A6 is distributed in the colon. In contrast, UT-B is widely distributed and is expressed in multiple tissues and organs such as the brain, heart, spleen, intestine, kidney, bladder, testis and vascular endothelium.

**Figure 3 biomedicines-14-00031-f003:**
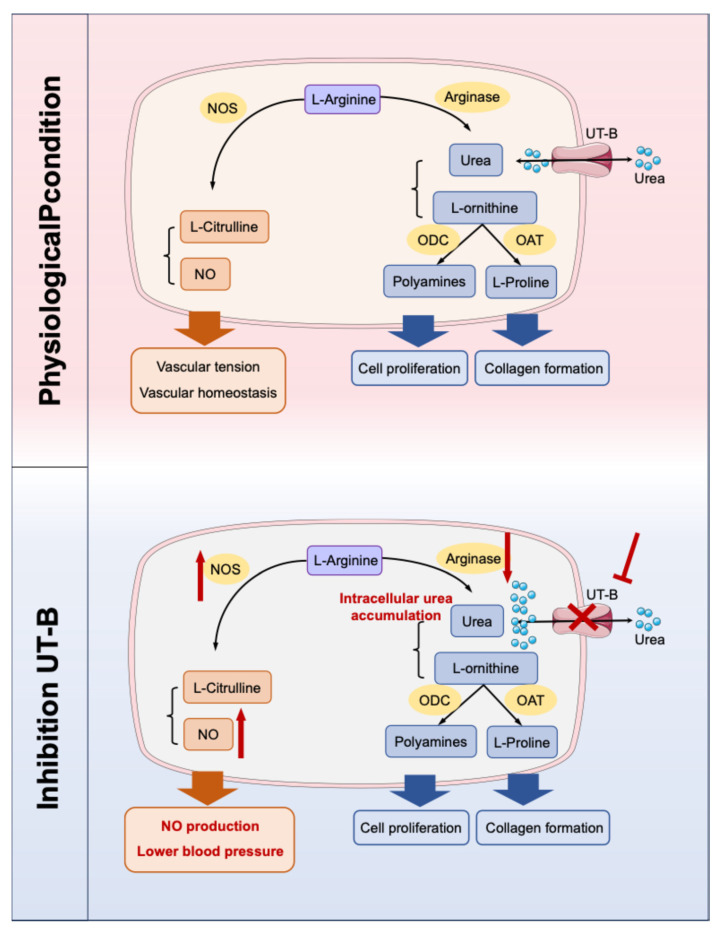
Two competing metabolic pathways for L-arginine. When UT-B is inhibited, the concentration of urea in endothelial cells increases. Excessive urea can down-regulate arginase, thereby reducing the substrate L-arginine–arginase metabolic pathway and promoting the biosynthesis pathway of NO. NO: nitric oxide; NOS: NO synthase; UT-B: urea transporter-B; ODC: ornithine decarboxylase; OAT: ornithine aminotransferase. The red sharp arrow denotes promotion/activation, while the red blunt arrow represents inhibition/blockade.

**Table 1 biomedicines-14-00031-t001:** Underlying possible roles of UTs in the cardiovascular diseases.

Cardiovascular Diseases	UTs	Location	Study Type	Model/Species	Key Observation	Mechanistic Insight
**Hypertension**	UT-B	Vascular endothelial cells	Basic Physiological Study; Basic Mechanistic Study	UT-B knockout mice; spontaneously hypertensive rats	UT-B inhibition can induce vasodilation	Inhibition of the arginase and increase of the biosynthesis of NO
	UT-A1	Apical membrane of IMCD	Basic Mechanistic Study; Genome-wide association studies (GWAS)	Angiotensin II-induced hypertension rats; UT-A1/3 knockout mice; Cohort of 405 subjects received nifedipine	UT-A1 inhibition induces osmotic diuresis	Block of urea reabsorption in the medullary collecting ducts
**Uremic Cardiomyopathy**	UT-A	Cardiomyocytes and cardiac fibroblasts	Basic Mechanistic Study	Uremic cardiomyopathy mice; UT-A1/3 knockout mice; CKD mice	High urea triggers myocardial remodeling and fibrosis	Modulation of intracardiac RAS activity
**Myocardial Hypertrophy and Heart Failure**	UT-A	Heart	Basic Physiological Study	uremia rats; hypertensive rats; human hearts with dilated cardiomyopathy	UT-A proteins increase in the end-stage heart failure	Possible important biomarkers
	UT-B	Heart	Basic Physiological Study	UT-B knockout mice	UT-B deficiency induces myocardial hypertrophy and cardiac remodeling	Damage in cardiac adaptation to stress overload, mitochondrial function, oxidative stress
**Cardiac Conduction Disorders**	UT-B	Cardiomyocytes	Basic Physiological Study	UT-B knockout mice	Urea accumulation caused by UT-B deficiency may promote atrioventricular block	Impairment in myocardial excitability and conductivity and mitochondrial dysfunction, Inflammation, oxidative stress
**Atherosclerosis and Endothelial Dysfunction**	UT-B	Vascular endothelial cells	Basic Physiological Study	UT-B knockout mice	High urea accelerats foam cell formation and atherosclerotic plaques	Carbamylation modification of proteins such as carbamylated low-density lipoprotein (cLDL)
					Pathological increase in urea may reduce the synthesis of NO	Inhibition of the transport of L-arginine

UT: urea transporter; NO: nitric oxide; IMCD: inner medullary collecting duct; CKD: Chronic Kidney Disease; RAS: renin-angiotensin system.

## Data Availability

No new data were created or analyzed in this study. Data sharing is not applicable to this article.

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
