# Peer review of "The Physiological Roles and Pathological Implications of Urea Transporters in the Cardiovascular System"

_biomedicines, 2025, doi:10.3390/biomedicines14010031_

Round 1

Reviewer 1 Report

Comments and Suggestions for Authors

This review summarizes the distribution of urea transporters (UTs) in the cardiovascular system, explores their potential roles in cardiovascular pathophysiology, and discusses the challenges associated with targeting UTs as a novel therapeutic approach for cardiovascular diseases. UTs may serve as modulators of cardiovascular disease and as cardioprotective therapeutic targets. The topic is both interesting and significant, and the conclusion effectively encapsulates the main objectives. I recommend this manuscript for publication. However, there are some shortcomings that require further optimization and improvement. Below are more detailed comments:

  1. Although the content is clear, I recommend revising the structural features of UTs. Consider replacing some summarized characteristics with images, as visuals can significantly enhance the manuscript's overall engagement. The 3D structure, amino acid domains, and both existing and potential drug-binding sites of UTs should be incorporated.

  1. In the Introduction section, discussing amino acid metabolism as the primary site of urea production, along with the diffusion of urea into the cardiovascular system, may enhance the coherence and overall integrity of the Introduction.

  1. Please cite more relevant literature to emphasize the roles of UTs in the mechanism of urine concentration within the cardiovascular system. Additionally, summarize the effects of urine on the cardiovascular system.

Author Response

  1. Although the content is clear, I recommend revising the structural features of UTs.  Consider replacing some summarized characteristics with images, as visuals can significantly enhance the manuscript's overall engagement. The 3D structure, amino acid domains, and both existing and potential drug-binding sites of UTs should be incorporated.

Response: Thank you for your helpful comments. We have revised the structural features of UTs and have added structural representation for illustration.

  1. In the Introduction section, discussing amino acid metabolism as the primary site of urea production, along with the diffusion of urea into the cardiovascular system, may enhance the coherence and overall integrity of the Introduction.

Response: We appreciate the suggestion. We have revised the Introduction section accordingly to enhance the coherence and overall integrity.

  1. Please cite more relevant literature to emphasize the roles of UTs in the mechanism of urine concentration within the cardiovascular system. Additionally, summarize the effects of urine on the cardiovascular system.

Response: Thank you for the comments. We have revised both the Introduction and Discussion sections accordingly.

Reviewer 2 Report

Comments and Suggestions for Authors

This comprehensive review systematically examines the emerging significance of urea transporters (UTs) beyond their traditional renal functions, with particular focus on their novel roles in cardiovascular pathophysiology. The overall framework of the review is reasonable, but in terms of the depth of discussion and logical coherence in key sections,the following points should be addressed:

  1. Urea mainly regulates the urine concentration function of the kidney by modulation of osmotic pressure. Then, is the impact of urea on other organ systems also related to its osmotic pressure regulation function? It is suggested to revise it and systematically introduce the physiological functions of urea.
  2. The fourth paragraph of the Introduction mentions urea as a clinical marker for evaluating renal function, which is not logically coherent with the previous text. It is suggested to make revisions.
  3. The current content mentions that UT-B is widely expressed, but it does not specify the exact tissue source of UT-B that regulates the progression of hypertension, such as vascular endothelium or kidney. It is suggested to discuss the contribution of UT-B with different tissue specificity and its possible mechanismin blood pressure regulation.
  4. Is the regulatory effect of UT-A1/A3 on uremic vascular disease only through mediating urea permeability? UTs are permeable not only to urea but also to water and urea analogues. It is suggested to supplement the discussion on the toxic effects of these permeants and whether UT-A1/A3 mediates the disease process through multiple factor interactionsï¼›
  5. In the part ofMyocardial Hypertrophy and Heart Failure, UT regulates the expression of ODC, suggesting that it may affect polyamine metabolism. It is suggested that the author search the literature to explore the association between UT and polyamine metabolic pathways, such as the role of polyamines in cell proliferation and hypertrophy.

Author Response

  1. Urea mainly regulates the urine concentration function of the kidney by modulation of osmotic pressure.Then, is the impact of urea on other organ systems also related to its osmotic pressure regulation function? It is suggested to revise it and systematically introduce the physiological functions of urea.

Response: We agree. In response to your comment, we have expanded the Discussion to include the role of urea in regulating osmotic pressure within the cardiovascular system. Additionally, we have revised the Introduction section to include the physiological functions of urea.

  1. The fourth paragraph of the Introduction mentions urea as a clinical marker for evaluating renal function, which is not logically coherent with the previous text.   It is suggested to make revisions.

Response: We sincerely thank the reviewer for the insightful feedback. According to this comment, we have updated the Introduction section .

  1. The current content mentions that UT-B is widely expressed, but it does not specify the exact tissue source of UT-B that regulates the progression of hypertension, such as vascular endothelium or kidney. It is suggested to discuss the contribution of UT-B with different tissue specificity and its possible mechanismin blood pressure regulation.Is the regulatory effect of UT-A1/A3 on uremic vascular disease only through mediating urea permeability? UTs are permeable not only to urea but also to water and urea analogues. It is suggested to supplement the discussion on the toxic effects of these permeants and whether UT-A1/A3 mediates the disease process through multiple factor interactionsï¼›

Response: Thank you for the comments. We have revised both the Hypertension and Discussion sections accordingly to discuss the contribution of UT-B with different tissue specificity.

Regarding the permeability of UTs to other permeants, UT-B also transports water, ammonia, and various urea analogues (formamide, acetamide, methylurea, methylformamide, thiourea, acrylamide and carbonate) in addition to urea, while UT-A-mediated transport is relatively specific. We have revised both the Atherosclerosis and Endothelial Dysfunction and Discussion sections.

  1. In the part of Myocardial Hypertrophy and Heart Failure, UT regulates the expression of ODC, suggesting that it may affect polyamine metabolism. It is suggested that the author search the literature to explore the association between UT and polyamine metabolic pathways, such as the role of polyamines in cell proliferation and hypertrophy.

Response: Thank you for your suggestion. We have revised the relevant discussion in Myocardial Hypertrophy and Heart Failure section.

Reviewer 3 Report

Comments and Suggestions for Authors

I read the manuscript “The Physiological Roles and Pathological Implications of Urea Transporters in the Cardiovascular System” with a great interest. 

This manuscript provides a good overview of the emerging roles of urea transporters (UT-A and UT-B) in cardiovascular biology and disease. The discussed area is timely and relevant, as urea metabolism is increasingly recognized as more than just a marker of renal function, especially in the context of cardiorenal interactions.

The authors successfully took together evidence from molecular biology, animal models, and limited human studies to propose urea transporters as both mechanistic players and potential therapeutic targets in cardiovascular disorders, including hypertension, uremic cardiomyopathy, heart failure, conduction abnormalities, and atherosclerosis.

Overall, the manuscript is informative and well-organized, but there are several issues that should be addressed to strengthen its scientific rigor and clarity.

Strengths:

1) The review is organized by disease categories, which makes it easy to follow and clinically relevant.

2) The link between urea transport, cardiovascular pathology, and CKD-related complications is highly relevant and of interest to both basic and translational researchers.

3) The manuscript goes beyond descriptive biology and offers mechanistic explanations for how UTs may influence cardiovascular function.

4) The authors cite many recent studies (including 2023–2025 papers), showing that the review reflects the current state of the field.

5) The final section well acknowledges the reliance on animal models and the need for more human studies.

Major Points for Improvement:

1) In line 367, the authors describe this work as a systematic review. However, the manuscript does not currently follow systematic review standards. I strongly recommend that the authors:

Follow PRISMA (Preferred Reporting Items for Systematic Reviews and Meta-Analyses) guidelines.

Clearly describe the search strategy: databases used, keywords, time frame, and inclusion/exclusion criteria.

Include a PRISMA flow diagram showing article selection and screening steps.

Add a table summarizing included studies (study type, model, sample size, main findings).

2) Most mechanistic conclusions are based on knockout mouse studies. While these are valuable, the authors should be more careful not to overgeneralize to human disease and should strengthen the discussion of translational limitations.

3) There are sections (e.g., the UT-B–arginine–NO axis) where different mechanisms are presented but not fully reconciled. These conceptual contradictions should be better organized or explicitly discussed.

4) The figures are helpful but visually basic. Adding clearer labeling and improving graphical quality would increase their impact. The summary table would benefit from more structure, such as adding columns for study type, species, or strength of evidence.

5) The manuscript is understandable but often too complex in sentence structure. Simplifying some long sentences and removing repetitive phrasing would improve readability and overall flow.

Minor Suggestions:

Consider adding a short section on future directions and clinical translation challenges, especially regarding drug development targeting UTs.

Improve consistency when referring to specific UT isoforms and experimental models.

Double-check terminology and reduce overstatements where human evidence is limited.

Overall Assessment:

This is a solid and interesting review on an emerging topic in cardiovascular and renal biology. With improved methodological transparency (especially regarding its classification as a systematic review), clearer mechanistic discussion, and some polishing of figures and language, it could become a very strong contribution to the field.

At its current stage, I would recommend major revision before acceptance.

Author Response

  1. In line 367, the authors describe this work as a systematic review. However, the manuscript does not currently follow systematic review standards. I strongly recommend that the authors:Follow PRISMA (Preferred Reporting Items for Systematic Reviews and Meta-Analyses) guidelines. Clearly describe the search strategy: databases used, keywords, time frame, and inclusion/exclusion criteria. Include a PRISMA flow diagram showing article selection and screening steps. Add a table summarizing included studies (study type, model, sample size, main findings).

Response:We sincerely thank you for raising this important methodological point.  We have therefore corrected the term "systematic review" throughout the manuscript and strengthened the logical presentation of cited evidence to ensure that the study type and key findings supporting each major argument are clearly stated.

  1. Most mechanistic conclusions are based on knockout mouse studies. While these are valuable, the authors should be more careful not to overgeneralize to human disease and should strengthen the discussion of translational limitations.

Response:Thanks for your insightful suggestions. To address this, we have revised the descriptions of the mechanistic conclusions and strengthened the discussion on translational implications in the Summary section..

  1. There are sections (e.g., the UT-B-arginine-NO axis) where different mechanisms are presented but not fully reconciled. These conceptual contradictions should be better organized or explicitly discussed.

Response: We sincerely appreciate the constructive comments. We have incorporated relevant discussion in the Summary section for better integration of the mechanisms within the UT-B-arginine-NO axis.

  1. The figures are helpful but visually basic. Adding clearer labeling and improving graphical quality would increase their impact. The summary table would benefit from more structure, such as adding columns for study type, species, or strength of evidence.

Response: Thank you for the constructive suggestions. We have redrawn the figures and incorporated a structural representation of UTs for better illustration. Meanwhile, we have revised the summary table to include columns for 'Study Type' and 'Model/Species' to provide a more structured overview of the evidence.

  1. The manuscript is understandable but often too complex in sentence structure. Simplifying some long sentences and removing repetitive phrasing would improve readability and overall flow.

Response: Thank you for pointing out the overly verbose and repetitive sentences. We have revised the manuscript to simplify sentence structures and remove repetitive phrasing.

  1. Consider adding a short section on future directions and clinical translation challenges, especially regarding drug development targeting UTs.

Response: As suggested, we have added a short section to the Summary section.

  1. Improve consistency when referring to specific UT isoforms and experimental models.

Response: Thank you for the comments. We have revised the manuscript to ensure consistent terminology when referring to specific UT isoforms and experimental models throughout the text.

  1. Double-check terminology and reduce overstatements where human evidence is limited.

Response: Thank you for pointing out. We have carefully reviewed to ensure precise terminology and reduce overstatements.

Round 2

Reviewer 3 Report

Comments and Suggestions for Authors

The Authors responded to all my major/minor concerns. They also revised the paper accordingly - I don't have any further comments on it.